# A Microwave-Assisted Boudouard Reaction: A Highly Effective Reduction of the Greenhouse Gas CO_2_ to Useful CO Feedstock with Semi-Coke

**DOI:** 10.3390/molecules26061507

**Published:** 2021-03-10

**Authors:** Huan Dai, Hong Zhao, Siyuan Chen, Biao Jiang

**Affiliations:** 1Green Chemical Engineering Research Center, Shanghai Advanced Research Institute, Chinese Academy of Sciences, Shanghai 201210, China; daihuan2018@sari.ac.cn; 2University of Chinese Academy of Sciences, Beijing 100049, China; 3Shanghai Green Chemical Engineering Research Center, Shanghai Institute of Organic Chemistry, Chinese Academy of Sciences, Shanghai 200032, China; chensy@sioc.ac.cn

**Keywords:** microwave heating, CO_2_ conversion, semi-coke, Boudouard reaction

## Abstract

The conversion of CO_2_ into more synthetically flexible CO is an effective and potential method for CO_2_ remediation, utilization and carbon emission reduction. In this paper, the reaction of carbon-carbon dioxide (the Boudouard reaction) was performed in a microwave fixed bed reactor using semi-coke (SC) as both the microwave absorber and reactant and was systematically compared with that heated in a conventional thermal field. The effects of the heating source, SC particle size, CO_2_ flow rate and additives on CO_2_ conversion and CO output were investigated. By microwave heating (MWH), CO_2_ conversion reached more than 99% while by conventional heating (CH), the maximum conversion of CO_2_ was approximately 29% at 900 °C. Meanwhile, for the reaction with 5 wt% barium carbonate added as a promoter, the reaction temperature was significantly reduced to 750 °C with an almost quantitative conversion of CO_2_. Further kinetic calculations showed that the apparent activation energy of the reaction under microwave heating was 46.3 kJ/mol, which was only one-third of that observed under conventional heating. The microwave-assisted Boudouard reaction with catalytic barium carbonate is a promising method for carbon dioxide utilization.

## 1. Introduction

In the face of climate change due to global warming, converting carbon dioxide into synthetic, flexible and useful molecules rather than capturing CO_2_ for storage is the most attractive approach to climate change mitigation solutions (Scheme 1) [1]. Nevertheless, the dynamics and thermodynamic stability of CO_2_ severely limit conversion. Efficient reducing agents and precious metal catalysts are required to facilitate its conversion into useful chemicals [2]. A simple and well-known response is the Boudouard reaction (C + CO_2_ = 2CO) in which CO_2_ reacts with carbon to produce carbon monoxide [3]. CO is not only the feedstock for Fischer–Tropsch’s synthetic oils and waxes but also a raw material of carbonylation reactions to synthesize a number of fine chemicals [4] and it provides important hydrogen energy through water-gas shift reactions [5].

However, the Boudouard reaction is a highly endothermic gas-solid reaction with a large positive enthalpy (172 kJ/mol at 298 K) in which very high temperatures, typically above 700 °C, are required to shift the equilibrium towards CO production [3]. Over the past decade, interest in using microwaves to activate chemical reactions has become widespread, creating significant advantages in the field of heterogeneous catalysis [6]. In particular, high-temperature microwaves have a special advantage in inducing chemical reactions derived from heterogeneous catalysts with unique opportunities to control the energy input [7]. Therefore, the ideal microwave catalyst has a dual role both as a catalyst for chemical reactions and as an efficient converter of the thermal energy required for the event and reaction activation from microwave energy [8]. Carbonaceous materials are known as excellent microwave receptors [9] and the high temperatures required for the Boudouard reaction could be provided by microwave irradiation [10]. Microwave-enhanced CO_2_ gasification of oil palm shell char [11,12], coconut shell char [13] and sewage sludge waste has been studied [14]. The results showed that microwave radiation was more effective than conventional heating and the reaction temperature was reduced by approximately 200 °C in a short time. The reactivity of biochar was found to be dependent on the pore structure of the char and its morphology, the catalytic activity of the associated ash, the availability of carbon active sites as well as catalytic active sites on the char, the thermal history of the char during pyrolysis and the type of carbon source.

China is a coal-rich country and implementing carbon neutrality is the country’s strategy for mitigating climate change. Semi-coke, named blue coal, is prepared from Jurassic low-quality coal with reserves of 100 billion tons in China [15]. As a new type of carbon material with its high fixed carbon content, high chemical activity and low ash content, semi-coke can replace char (metallurgical coke) and is widely used in the chemical, smelting, gasification and other industries. In this paper, we report the results of an investigation specifically focused on the conversion of CO_2_ into CO with semi-coke under microwave irradiation and catalysis. To the best of our knowledge, similar data are not found in the literature and the Boudouard reaction was carried out with a CO_2_ conversion rate of 99% at 900 °C with an excellent microwave absorption performance of the semi-coke. When barium carbonate was added as a promoter, the reaction temperature was significantly reduced to 750 °C with an almost quantitative conversion of CO_2_.

## 2. Results and Discussions

### 2.1. Temperature Correction of the Microwave Fixed Bed Reactor

The application of microwave irradiation to a gas-solid reaction has been studied for more than 20 years. However, due to the practical difficulties of the temperature measurement and uniform temperature distribution in the microwave (MW) field, comparing reaction performances under conventional heating (CH) and microwave heating (MWH) always would be explained as the formation of hot spots in the reaction system as a consequence of MWH. In this paper, we obtained a relatively even temperature field in the microwave fixed bed by using three silicon carbide (SiC) rings with a width of 0.5 cm as thermal loads that were sheathed outside the quartz tube. Moreover, three optical radiation pyrometers were used to monitor the surface temperature of the bed. Before the Boudouard reaction experiments, we corrected the heating temperature and confirmed the constant temperature area in the microwave fixed bed reactor. We used a silicon carbide rod with a length of 13 cm and a diameter of 3.5 cm as a heat carrier for the temperature range correction (SiC is an excellent microwave absorber material [16,17]). Figure 1 shows the temperature distribution of the bed with an error of ±10 °C, identifying the uniformity of the temperature distribution. After that, we measured the temperature distribution using semi-coke (SC) as the heat carrier. The SC bed exhibited a relatively even temperature zone ranging from about ±3 cm from the center of the temperature zone, which met the needs of our experiments. The temperature distribution of the fixed bed is shown in Figure 1 and its error was within ±25 °C, which also met the needs of our experiments.

### 2.2. The Boudouard Reaction in Both Microwave and Conventional Thermal Fields

To investigate the characteristics of the CO_2_ gasification of SC in both microwave and conventional thermal fields, typical experiments were conducted with an SC particle size of 0.200–0.450 mm and a CO_2_ space velocity of 3.20 m^3^/(m^3^(SC)·h) at 750 °C. The effect of the temperature on the Boudouard reaction in both fields is shown in Figure 2a,b. As we can see from Figure 2a, the CO_2_ conversion and SC conversion by conventional heating increased slowly with an increase in temperature but were at a low level. CO_2_ conversion increased from 5.0% at 800 °C to 10.3%, 14.4% and 29.1% at 850 °C, 900 °C and 950 °C, respectively. The results were consistent with the strong endotherm of the Boudouard reaction. The high temperature pushed the balance to CO production when heated by conventional methods. Figure 2b shows the experimental results when heated by microwave irradiation. Clearly, when the reaction temperature was 750 °C, the CO_2_ conversion rate was 77.2%, which was much higher than that carried out at 950 °C by conventional heating. When the reaction temperature increased to 900 °C, the CO_2_ conversion increased to 99.7%. In addition, Figure 2c compares the CO output of the two heating sources. From Figure 2c, it can be clearly seen that by conventional heating, the CO output ranged from 8.1 mmol at 800 °C to 59.1 mmol at 950 °C while by microwave heating the CO output increased from 1163 mmol at 750 °C to 2000 mmol at 900 °C, which was more than 60 times than that obtained by conventional heating. The clear CO output at 950 °C was only approximately one-twentieth of that carried out in the microwave reactor at 750 °C. We know that SC is a good heating conductor and the possible temperature gradient derived from the characteristics of microwave heating would not be great enough to cause such a distinct Boudouard reaction performance between the two thermal fields. Therefore, the obviously higher CO_2_ conversion and CO output obtained in the microwave reactor were not caused by incorrect temperature measurements or uneven temperature distributions in the microwave reactor. The fact that the Boudouard reaction under microwave heating proceeded more easily suggested a great enhancement of the Boudouard reaction with microwave heating.

This superior performance of microwaves is believed to be related to the mechanism of microwave heating [18]. SC is an excellent material for absorbing microwaves that can be firmly coupled with microwaves to carry out chemical reactions. When the microwave acted on the SC, the microwave energy could directly couple with carbon molecules so that the dielectric heating effect of the microwave could effectively heat the raw material. This is different from conventional heating in that the main mechanism of microwave absorption in dielectric materials is to excite electronic oscillations at the frequency of the microwave source.

### 2.3. Influence of the SC Particle Size

The effect of the SC particle size on CO_2_ conversion and CO output in the microwave reactor was investigated at a temperature of 800 °C and a CO_2_ flow rate of 125 mL/min. As shown in Figure 3a, the CO_2_ conversion improved from 83.0% for particles in the range of 0.850–1.400 mm to 98.0% for particles in the range of 0.200–0.450 mm; at the same time, the CO_2_ conversion was maintained above 90%. As the particle size of the SC decreased, the gasification reactivity improved. This is because in the gasification process, the smaller particles of SC have better heat and mass transfer processes. Within a certain particle size range, the decreased SC particle size would give a larger contact area between CO_2_ and SC, causing a more complete reaction. In addition, the reduced SC particle size was helpful in reducing the effects of internal diffusion, which could also increase the gasification reaction rate. It can be seen that the CO_2_ conversion of the SC particle size < 0.200 mm was generally lower than that of the SC particle size 0.200–0.450 mm as the reaction time increased and this may be because too small a particle size would give a larger external diffusion resistance and the CO_2_ reactant would be more difficult to evenly diffuse into the SC bed. Figure 3b shows the effect of different particle sizes on the CO output. We can see that the CO output also shows the same trend as the CO_2_ conversion. When the SC particle size was 0.200–0.450 mm, the maximum CO output was 1371 mmol.

### 2.4. Influence of the CO_2_ Gas Flow Rate

Figure 3c shows the influence of the CO_2_ gas flow rate on the CO_2_ conversion of SC. The CO_2_ conversion decreased from 97.3% to 67.2% when the CO_2_ flow increased from 125 mL/min to 200 mL/min. This was because the increase in the CO_2_ flow shortened the residence time of CO_2_ gas in the carbon bed and then reduced the chance of CO_2_ conversion therefore causing a significant drop in CO_2_ conversion. Figure 3d shows the CO output under the different gas flows. The results indicated that the CO output increased from 1462 mmol to 1581 mmol when the CO_2_ flow increased from 125 mL/min to 200 mL/min. However, with the increase in CO_2_ flow, the output of CO was not obviously changed. This might be due to the increase in CO_2_ flow, resulting in the reduction in the CO_2_ external diffusion resistance and the increase in CO_2_ diffusion into SC particles, which intensified the CO_2_ conversion reaction.

### 2.5. Influence of the Catalyst

It has been reported that the salts of alkali metals and alkaline earth metals such as Na, K and Ca have a good catalytic performance for char gasification reactions [19,20,21] but the effect of barium salt on the Boudouard reaction has been scarcely investigated. Here, BaCO_3_ was used as a catalyst and compared with CaO in the gasification reaction of SC and CO_2_ under microwave radiation. Figure 4a exhibits the CO_2_ conversion of the SC with/without 5 *wt%* CaO and a BaCO_3_ addition at a reaction temperature of 750 °C and a CO_2_ flow rate of 175 mL/min under microwave heating. Without a catalyst, the initial CO_2_ conversion at 750 °C was 86.2% and then decreased quickly to approximately 40% after 144 min. Under the catalysis of BaCO_3_ and CaO, the initial CO_2_ conversion was 99.9% but BaCO_3_ exhibited a better catalytic life. Figure 4b shows that the CO output also increased significantly from 1689 mmol to 2392 mmol and 2520 mmol when CaO and BaCO_3_ were used as catalysts. The experimental results indicated that BaCO_3_ and CaO had an excellent catalytic performance for the Boudouard reaction and reduced the reaction temperature. The reason for the higher reactivity of SC catalyzed by BaCO_3_ would be that BaCO_3_ improved the adsorption capacity of SC for CO_2_, thereby obtaining a higher reactivity.

Figure 5 shows the surface morphologies of gasified SC before and after a reaction under different reaction conditions. It can be seen from Figure 5a that the un-gasified SC had a relatively smooth surface and the carbon layer structure was complete and without obvious pores inside the bulk SC particles. By conventional heating, the SC particles shown in Figure 5b exhibited an inconspicuous lamellar structure with a few large pores and the carbon layer structure was relatively complete. The possible reason is that gasified SC hardly occurs at low temperatures in conventional heating. The morphology of gasified SC under microwave radiation is shown in Figure 5c. The surface of the SC was obviously damaged and a few large pores were formed. As shown in Figure 5d, BaCO_3_ particles on gasified SC obtained at 750 °C showed that the carbon layer structure was further destroyed and the surface carbon structure was more uniformly distributed. The results indicated that the reaction rate of CO_2_ on the surface of SC was more intense in the presence of barium salt. Table 1 shows the specific surface areas and volume pores for SCs heated by different techniques. The specific surface area and pore volume of raw SC were 2.16 m^2^/g and 1.30 × 10^−3^ cm^3^/g, respectively. After conventional and microwave heating, the specific surface area was increased from 2.16 m^2^/g to 3.60 m^2^/g for CH-SC, 18.2 m^2^/g for MW-SC and 14.3 m^2^/g for MW-SC/BaCO_3_ while the pore volume also showed the same trend. The results indicated that microwaves enhanced the reactivity of SC, resulting in a higher CO_2_ conversion and CO output.

### 2.6. Analysis of Gasification Kinetics

In order to compare the reaction performances in both thermal fields, the gasification kinetics in the two thermal fields were studied based on the experimental data exhibited in Section 2.2. Both diffusional and mass transfer influences were ruled out. We know that the volume reaction model is a commonly used kinetic model in the Boudouard reaction. The volume reaction model (VRM) assumes that the gasification reaction occurs at the active site and is evenly distributed throughout the carbon particles [22,23,24,25]. By the kinetic analysis of the isothermal gasification experiment and then the conversion of the first-order kinetics into the Arrhenius relationship, the kinetics of several coal conversion processes could be represented. Therefore, we performed kinetic calculations on the experimental results under conventional and microwave (non-catalysis) thermal fields. The reaction rate equation was expressed as follows:(1)$$r=\frac{dX}{dt}=k_{VRM}\left(1-X\right).$$
(2)$$\ln\left(1-X\right)=-k_{VRM}t.$$

From the graph of ln(1−x)−t, the value of the first-order reaction rate constant $k_{VAM}$ at each temperature could be obtained. The temperature dependence of $k_{VAM}$ was expressed by the Arrhenius equation as follows:(3)$$k_{VRM}=Aexp\left(-\frac{E_{a}}{RT}\right).$$
(4)$$lnk_{VRM}=-\frac{E_{a}}{RT}+lnA.$$

In the formula, the $k_{VRM}$ is the rate constant, min^−1^; the A is the pre-exponential factor, min^−1^; the $E_{a}$ is the reaction activation energy, kJ/mol; the R is the gas constant (8.314 J/mol/K) and the X is the SC conversion.

Figure 6a,b shows the relationship between the microwave and conventional reaction times and carbon conversions. The slope of the fitted straight line was the reaction rate constant at this temperature. It was suggested that the reaction rate constant increased with an increase in temperature in the two heating methods, which indicated that when the temperature increased, the reaction rate increased. Figure 6c shows the Arrhenius diagram of microwave and conventional reactions. For microwave and conventional reaction experiments, higher regression coefficients were obtained. The activation energy and pre-exponential factor were obtained from the slope of the fitted straight line. Table 2 shows the activation energy and pre-exponential energy under microwave and conventional heating. The activation energies were 47.1 kJ·mol^−1^ and 148.9 kJ·mol^−1^, respectively. The activation energy was much smaller than that under conventional heating. The large difference between the microwave-induced reaction rate and the thermal reaction rate indicated that the rate constant must have had a fundamentally different dependence on temperature, which was demonstrated in the Arrhenius parameter. In addition, as shown in Table 2, the exponential factor of the microwave heating reaction was significantly smaller than that of the conventional thermal reaction by more than 1000 orders of magnitude. Hunt et al. [26] suggested that a lower reaction temperature in the microwave experiment caused a lower collision frequency of reactants to reduce the pre-exponential term. That is to say, microwave heating should lead to a more efficient collision frequency of reactants.

## 3. Materials and Methods

### 3.1. Materials

The semi-coke was provided by the Shanxi Lu’an Coal Industry (Changzhi, China). The ultimate analysis and proximate analysis are shown in Table 3. The semi-coke was dried, crushed and sieved into four particle sizes of <0.200 mm, 0.200–0.450 mm, 0.450–0.850 mm and 0.850–1.40 mm. Before the experiment, the semi-coke was continuously dried at 105 °C for 12 hrs and then stored in a well-sealed dryer for use in the experiment. This article named semi-coke as SC. Ar and CO_2_ (purity of 99.9%), calcium oxide (CaO, 99% from Aldrich) and barium carbonate (BaCO_3_, 99% from Aldrich) were used in the experiment.

### 3.2. Methods

The reactions heated by microwaves were carried out in a microwave fixed bed reactor (Synotherm Corporation, Changsha, China) and the experimental schematic diagram is shown in Appendix A. The microwave fixed bed reactor was heated by a monomodal cavity with a magnetron generator operating at 2.45 GHz (the maximum output power was 1500 W). The reactor was equipped with a temperature automatic control system and a quartz glass tube with an inner diameter of 38 mm and a length of 90 cm. For conventional heating, the reactions were carried out in a fixed bed reactor under the same conditions. In the experiment, approximately 40.0 ± 0.01 g of SC was first put into a quartz glass tube. Before the reaction, Ar was introduced into the reactor at a flow rate of 100 mL/min for 30 min. When the temperature reached the desired value, CO_2_ gas was precisely introduced into the reactor and the reaction started. The composition of the gas products was analyzed online by a gas chromatograph (GC) (SHIMADZU, 2014C, Japan) equipped with a thermal conductivity detector (TCD) and a packed column (Porapak Q 80–100 mesh, 2 m × 4 mm) and the reacted SC was analyzed after the reaction. In our experiments, the heating rate in the conventional fixed bed reactor was 10 °C/min while in the microwave reactor the heating rate was 60 °C/min although the input power of the conventional fixed bed reactor and the microwave oven were 1200 kw and 800 kw, respectively.

### 3.3. Carbon Characterization and Data Analysis

The pore structures of SC were measured using a Micromeritics ASAP 2460 N_2_ adsorption analyzer (ASAP 2460, Micromeritics, the United States). SC (0.1 g) was weighed and placed in a quartz tube for high-temperature vacuum degassing and then placed in a bottle containing liquid nitrogen at −196 °C for the N_2_ adsorption/desorption experiments. The specific surface area was calculated using the BET equation and the average pore size data were calculated from the H-K equation (Horvath-Kawazoe equation). The surface structure of SC was observed by scanning electron microscopy (SEM) (FEI Nova Nono 450, Netherlands) with an accelerating voltage of 5.0 kV.

The real-time composition of the outlet gas was measured by a GC and the CO_2_ conversion, CO output and SC conversion were calculated according to the changes before and after the CO_2_ reaction as follows:(5)$$Con_{SC}=\frac{m-m_{t}}{m-m_{ash}}$$
where m, $m_{t}$ and $m_{ash}$ represented the initial mass of SC, the mass of SC after the reaction and the mass of ash in SC, respectively.
(6)$$Con_{CO_{2}}=\frac{n_{b}-n_{a}}{n_{b}}$$
where $n_{b}$ and $n_{a}$ represented the moles of CO_2_ before the reaction and the moles of CO_2_ after the reaction, respectively.

## 4. Conclusions

The CO_2_ gasification of semi-coke was studied through microwave radiation. We found that microwaves do have an excellent enhancement effect on the reaction. The experimental results showed that the maximum CO_2_ conversion was 99.9% in the presence of barium carbonate. Based on the kinetic study, the activation energy for microwave heating was 47.1 kJ/mol while this value was 148.9 kJ/mol for conventional heating, indicating that microwaves could increase the Boudouard reaction and reduce the reaction activation energy. Thus, BaCO_3_ for an enhanced microwave chemical Boudouard reaction makes the zero emission of CO_2_ and the clean utilization of coal resources possible.

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
