# Peer review of "A Microwave-Assisted Boudouard Reaction: A Highly Effective Reduction of the Greenhouse Gas CO2 to Useful CO Feedstock with Semi-Coke"

_molecules, 2021, doi:10.3390/molecules26061507_

Round 1

Reviewer 1 Report

This manuscript deals with the reaction between CO2 and semi-coke to CO using either thermal or microwave heating. Both procedures are compared under different conditions. It is interesting but it needs to be revised taking into account the following issues:

- Please, give the conversion of semi-coke. Equation (5) indicates how XSC was calculated but only XCO2 is provided in the manuscript.

- It would be interesting for the reader to have the energy inputs for both thermal and microwave activations.

- Page 3, line 110. Fig. 4c should be replaced by Fig. 2c.

- Please, indicate reaction time for those graphs where CO production is given.

- The authors measured temperature distribution in the reactor bed before CO2 feeding. Is the temperature maintained under reaction conditions with CO2?

- Fig. 3a indicates that activity decreases with reaction time for particles <0.200 mm but not for particles in the range 0.200-0.450 mm. The authors explain it due to a larger external diffusion resistance for particles with smaller size. However, why does this effect occur after 50 min and not from the beginning? Have the authors checked the SC particle size after the reaction to check if attrition happened during the reaction? Have they observe pressure changes in the reactor?

- Page 6, line 205. The authors state “enhance the reality of SC”. What do they mean? According to Table 1, it is clear that specific surface area increases with the extent of SC transformation.

- Page 7, line 230. Please, write A as pre-exponential factor.

- The authors should justify the validity of the kinetics approach. Apparently, mass and energy transfer under the reaction conditions studied (very high conversions) in this manuscript can be rather limiting. Also, the authors comments on the existence of diffusion resistance in some conditions. The studied reaction involves gas-solid contact and so diffusional and mass transfer phenomena must be ruled out.

- Please, indicate the meaning of FC, M and V in proximate analysis in Table 3.

Reviewer 2 Report

Huan and co-workers demonstrated microwave assisted Boudouard reaction towards CO production using semi-coke. The influence of semi-coke particle size, CO2 gas flow rate, and catalyst has been reported. The paper demonstrates efficient production of CO from semi-coke and the greenhouse gas, CO2. However, it is reported (ref. 26) that the microwave reaction shifts the thermodynamics favoring the formation of CO and the reaction efficiency was demonstrated with charcoal. Thus, novelty of the work is demonstration of Boudouard reaction using Semi-coke.

I do not see supporting information; can’t see the Figure and table S1s

Revise title 

Round 2

Reviewer 1 Report

This manuscript is now acceptable for publication in Molecules.

Author Response

Thank you very much!